# Evaluation of Anti-Inflammatory and Antioxidant Effectsof Chrysanthemum Stem and Leaf Extract on Zebrafish Inflammatory Bowel Disease Model

**DOI:** 10.3390/molecules27072114

**Published:** 2022-03-25

**Authors:** Yi Li, Xia-Jin Liu, Shu-Lan Su, Hui Yan, Sheng Guo, Da-Wei Qian, Jin-Ao Duan

**Affiliations:** National and Local Collaborative Engineering Center of Chinese Medicinal Resources Industrialization and Formulae Innovative Medicine, Jiangsu Collaborative Innovation Center of Chinese Medicinal Resources Industrialization, Jiangsu Key Laboratory for High Technology Research of TCM Formulae, Nanjing University of Chinese Medicine, Nanjing 210023, China; liyi9823@163.com (Y.L.); xiajinliu1996@163.com (X.-J.L.); glory-yan@163.com (H.Y.); guosheng@njucm.edu.cn (S.G.); qiandwnj@126.com (D.-W.Q.)

**Keywords:** Flos Chrysanthemi, chrysanthemum stem and leaf, inflammatory bowel disease (IBD), zebrafish, UPLC-TQ/MS

## Abstract

Present studies have shown that Flos Chrysanthemi has anti-inflammatory and other effects and regulates intestinal function, while the chrysanthemum stem and leaf as non-medicinal parts of chrysanthemum have similar chemical components with chrysanthemum, but the activity and mechanisms are rarely elucidated. Therefore, this study used a DSS-induced zebrafish inflammatory bowel disease model to study the anti-inflammatory and antioxidant effects of chrysanthemum stem and leaf extracts. The results indicate that DSS induction leads to increased secretion of acidic mucin in the intestines of juvenile fish, enlargement of the intestinal lumen and the emergence of intestinal inflammation. Compared with the model group, each administration group differentially inhibited the expression of IL-1β, IL-8 and MMP9 in DSS-induced zebrafish, while upregulating the activity of superoxide dismutase. The quantitative analysis results showed that the flavonoids (including Linarin, Diosmetin-7-glucoside, Tilianin, etc.) and phenolic acids (including Isochlorogenic acid C, Isochlorogenic acid A, 1,3-Dicaffeoylquinic acid, etc.) in the alcohol extract were closely related with both anti-inflammatory and antioxidant activity, while the polysaccharides were also shown a certain anti-inflammatory and antioxidant activity. In conclusion, this study suggests that the flavonoids, phenolic acids and polysaccharides from chrysanthemum stem and leaf extracts can improve inflammatory bowel disease of zebrafish by regulating the expressions of IL-1β, IL-8 and MMP9.

## 1. Introduction

Inflammatory bowel disease (IBD) includes Crohn’s disease (CD) and ulcerative colitis (UC). Symptoms can include abdominal pain, vomiting, diarrhea, and weight loss [1,2,3]. With the incidence and prevalence of IBD increasing over time in different parts of the world, the disease has now become a global disease [4,5,6]. Some studies show that about 6.8 million people worldwide suffer from IBD [7]. In recent years, the pathogenesis of IBD has made some progress, involving environmental factors, psychological factors, genetic factors and intestinal microbes [8,9], but the specific pathogenesis is not clear. The main manifestations of the patient are abdominal pain, diarrhea, fever, weight loss, and possible sub-occlusion (spasm pain, vomiting, difficulty with feces/discharge) or other complications (i.e., abscess, fistula) [10]. At present, western medicine treatment drugs and means have been more common, such as corticosteroids, aminosalicylic acid, sulfasalazine, antibiotics [11] and immunosuppression [12]. These drugs can significantly reduce inflammation and improve clinical symptoms, but long-term use still has certain side effects [13,14], and the price of drugs is expensive, greatly reducing the patient’s life quality and satisfaction. In recent years, there have been some new preparations and therapies, such as biological preparations [15] and fecal bacteria transplantation [16], but no breakthrough has been made. Therefore, looking for a safe and effective treatment is particularly important. Traditional Chinese medicine (TCM) has long been used to treat many clinical diseases, such as diarrhea, infection and functional disorders [17]. TCM emphasizes the treatment based on syndrome differentiation, and TCM has fewer side effects and significant efficacy; thus, it is increasingly used in clinical practice [18].

Flos Chrysanthemi (chrysanthemum) is the dry flower head of *Chrysanthemum morifolium* Ramat. As a kind of Traditional Chinese medicine (TCM), it has been widely used in wind-heat cold, headaches and dizziness, eye swelling and pain, eye coma, sore carbuncle and toxins. The pharmacological effects of modern studies include antibacterial, anti-inflammatory [19], antioxidant, as well as regulation of intestinal dysfunction, which is a good alternative to antibiotics. Studies have also shown that chrysanthemum polysaccharide can improve colitis and functional constipation in rats by regulating intestinal microbes [20,21,22,23]. While the chrysanthemum stem and leaf contain the main chemical components that are similar to chrysanthemum, including flavonoids, polysaccharides, phenolic acids, volatile oils and terpenoids [24,25,26], it is reported that the chrysanthemum stem regulates gastrointestinal function [27,28]. However, anti-inflammatory and antioxidant effects and the mechanisms of the chrysanthemum stem and leaf are rarely elucidated.

Zebrafish (*Danio rerio*) is a tropical ornamental fish belonging to the Genus Danil of cyprinidae. The adult fish of zebrafish is small and easy to raise and has rapid development, short sexual maturity, strong reproduction, and transparent embryo development in vitro, which makes it easy to observe. These characteristics make zebrafish widely used in high-throughput screening. Since the 1980s, zebrafish has received attention from the field of genetics and biology, and has gradually become one of the important model vertebrates [29]. With the completion of zebrafish gene sequencing, the zebrafish genome sequence and human gene are conserved as high as 70% [30], which means that zebrafish experimental results can also be applied to humans; thus, zebrafish is increasingly widely used in the study of human diseases.

Therefore, in this study, a sodium dextran sulphate (DSS)-induced inflammatory bowel disease (IBD) model of zebrafish was established, and the intervention effect of chrysanthemum stem and leaf extracts was evaluated. The contents of total phenolic acids and total flavonoids in chrysanthemum stem and leaf were determined by ULTRA-performance liquid chromatography–triple quadrupole electrospray tandem mass spectrometry (UPLC–TQ/MS). Neutral and acidic polysaccharides were determined by the phenol–sulfuric acid method [31] and carbazol–sulfuric acid method [32]. This study will provide a useful reference for the exploration and utilization of chrysanthemum stem and leaf.

## 2. Results

### 2.1. Pathological Changes in DSS-Induced Enterocolitis

#### 2.1.1. Alcian Blue Staining

Whole mount alcian blue staining is a useful technique for visualizing the acidic mucin produced by zebrafish goblet cells [33]. Studies have shown that DSS induction leads to the accumulation of acidic mucin in the gut bulb [34]. The staining results showed that the intestinal fraction of DSS-induced zebrafish larvae was significantly stained blue compared to the control group, indicating the presence of acidic mucin accumulation (Figure 1).

#### 2.1.2. Hematoxylin Eosin Staining

By staining the pathological sections and placing them under the microscope and counting the proportion of the intestinal lumen area, it was seen that the intestinal tissue in the blank group did not show any obvious lesions. More serious intestinal epithelium damage was seen in the intestinal lumen after 72 h of 0.5% DSS modeling, with an increase in cupped cells and an enlargement of the intestinal lumen area compared to the blank group. The degree of intestinal epithelium shedding was reduced in the water extract administration group and alcohol extract administration group of chrysanthemum stem leaves compared with the model group, and some of the administration groups could reduce the enlargement of intestinal lumen caused by DSS modeling. The histological scores of the model group were significantly higher than those of the control group, and a comparison of the administration groups revealed a more pronounced reduction in the degree of pathology in S0.2, S0.6, C0.02 and C0.04. The HE staining results, histological score and percentage of intestinal lumen area are shown in Figure 2A–C.

### 2.2. Changes in Superoxide Dismutase Levels during DSS Induced Inflammation

The model was made by 0.5% DSS, and the concentration of 0.02, 0.04, 0.06 mg/mL alcohol extract of chrysanthemum stem and leaf and 0.1, 0.2, 0.4 mg/mL water extract of chrysanthemum stem and leaf were given to determine the corresponding SOD values. SOD activity in the model group was significantly lower than that in the blank group (*p* < 0.01). Compared with the model group, SOD activity in the alcohol extract groups of 0.02, 0.04, and 0.06 mg/mL and the water extract groups of 0.1, 0.2, and 0.4 mg/mL of chrysanthemum stem and leaf increased with the increase in drug concentration. There was no significant difference between the 0.1 mg/mL water extract administration group and the model group, while there were significant or extremely significant differences between the other groups and the model group, and the increase in SOD in the 0.06 mg/mL alcohol extract group and 0.4 mg/mL water extract group was close to that in the blank group, as shown in Figure 3.

It was concluded that 0.02, 0.04, and 0.06 mg/mL alcohol extract groups and 0.2 and 0.4 mg/mL water extract groups of chrysanthemum stem and leaf effectively inhibited intestinal injury of zebrafish and played a certain therapeutic effect on intestinal dysfunction induced by DSS. In conclusion, both alcohol and water extracts from chrysanthemum stem and leaf can inhibit oxidative damage of intestinal dysfunction zebrafish by changing the activity of superoxide dismutase (SOD), and they play a certain role in regulating intestinal dysfunction.

### 2.3. Inflammatory Cytokine Production in Larvae Exposed to DSS

The proinflammatory cytokines IL-1*β*, IL-8, and MMP9 were selected in this study; we then measured the expression levels of IL-1*β*, IL-8, and MMP9 using q-PCR. As shown in Figure 4, compared with the blank group, mRNA expressions of IL-1*β*, IL8, and MMP9 genes in the model group were significantly increased (*p* < 0.001), indicating that there was inflammation in the model group. Compared with the model group, the up-regulation of IL-1*β*, IL-8, and MMP9 induced by DSS was significantly inhibited in the water extract group and alcohol extract group (*p* < 0.05, 0.01, 0.001), and the inhibition effect of S0.2 in the water extract group and C0.04 in alcohol extract group was the most significant (*p* < 0.001). In conclusion, these results suggest that water extract from chrysanthemum stem and leaf and alcohol extract from chrysanthemum stem and leaf have a protective effect on DSS induced inflammatory bowel disease model of zebrafish and inhibit the DSS-induced inflammatory response.

### 2.4. Flavonoids and Phenolic Acids Were Determined by UPLC-TQ/MS Method

#### 2.4.1. Linearity

The results showed that the 19 flavonoids and phenolic acids showed a good linear relationship in the determination range. The regression equation, R^2^ and linear range of 19 reference substances were given in Table 1. The TQ chromatograms of the components to be measured in the mixed standards and the components to be measured in the samples are shown in Figure 5.

#### 2.4.2. Precision, Repeatability, and Stability

The precision test results showed that the RSD values of 19 phenolic acids and flavonoids were all less than 4.27%, indicating good precision of the instrument. Repeatability test results showed that the repeatability of 19 components in water extract (ST) and alcohol extract (CT) of chrysanthemum stem and leaf was less than 4.75% and 4.13%, respectively, indicating that the method had good repeatability. Stability test results showed that the stability of 19 components in water extract from chrysanthemum stem and leaf (ST) and alcohol extract from chrysanthemum stem and leaf (CT) of chrysanthemum stem and leaf was less than 4.50% and 4.95%, respectively, indicating that the tested solution had good stability within 24 h.

#### 2.4.3. Total Flavonoids, Total Phenolic Acid Content

Chrysanthemum leaf water extract and alcohol extract of the measured included Neochlorogenic acid, Chlorogenic acid, Caffeic acid, Cryptochlorogenic acid, 1,3-Dicaffeoylquinic acid, Isochlorogenic acid B, Isochlorogenic acid A, Isochlorogenic acid C, eight kinds of phenolic acids and Luteolin-7-*O*-*β*-D-glucoside, Apigenin-7-*O*-*β*-D-glucoside, Diosmetin-7-glucoside, Eriodictyol, Linarin, Tilianin, Apigenin, Diosmetin, Hesperetin, Acacetin, Luteolin and 11 kinds of flavonoids, a total of 19 components were detected.

The results showed that both water and alcohol extracts of chrysanthemum stem and leaf were rich in flavonoids and phenolic acids. The contents of total flavonoids and phenolic acid in water extract of chrysanthemum stem and leaf were 11.96 ± 0.96 mg/g. Total flavonoids and total phenolic acid were 5.95 ± 0.66 mg/g and 6.00 ± 0.59 mg/g, respectively. The total flavonoid and phenolic acid contents of the ethanol extracts from the stems and leaves of chrysanthemum were 26.91 ± 1.17 mg/g. The total flavonoid and phenolic acid contents were 18.25 ± 1.11 mg/g and 8.66 ± 0.68 mg/g, respectively. The 11 flavonoids in the alcohol extract of chrysanthemum stem and leaf were higher than those in the water extract of chrysanthemum stem and leaf, the content of Isochlorogenic acid A, B, and C were higher than that of water extract from chrysanthemum stem and leaf, but the content of total phenolic acid was higher. Figure 6 shows the content contrast of components in alcohol extract from chrysanthemum stem and leaf (CT) and water extract from chrysanthemum stem and leaf (ST) of chrysanthemum stem and leaf.

The neutral polysaccharide content in chrysanthemum stem and leaf was determined by the phenol–sulfuric acid method, and the acid polysaccharide content in chrysanthemum stem and leaf was determined by the carbazole–sulfuric acid method. The content of neutral polysaccharide was 74.36 ± 4.51 mg/g, acid polysaccharide was 26.72 ± 1.84 mg/g, and total polysaccharide was 101.08 ± 5.86 mg/g. The above results showed that neutral and acidic polysaccharides were important components of total polysaccharides in chrysanthemum stems and leaves, and the content of neutral polysaccharides was higher.

## 3. Discussion

As one of the important vertebrate animal models, zebrafish has the advantages of small size, transparent embryos, and a gastrointestinal system highly similar to humans, many digestive organs, such as the intestine, liver, gallbladder, and pancreas, are functionally similar to their mammalian counterparts, and have been increasingly used in the study of IBD and gastrointestinal diseases [35,36,37]. In this study, we first evaluated the anti-inflammatory and antioxidant effects of water and alcohol extracts from chrysanthemum stem and leaf using the zebrafish IBD model. The results showed that the DSS-induced zebrafish model had acidic mucin accumulation, enlarged intestinal lumen and broken epithelial cells in the intestinal fraction, while chrysanthemum stem and leaf extract inhibited the upregulation of various inflammatory factors (IL-1*β*, IL-8, MMP9) to varying degrees, showing its effective role in inflammatory regulation. Previously, TAO et al. [21,23] showed that chrysanthemum polysaccharide could improve ulcerative colitis by promoting the growth of beneficial intestinal flora, regulating intestinal microecological balance, and restoring the immune system, while significantly reducing the levels of inflammatory cytokines TNF-*α*, IL-6, IFN-*γ,* and IL-1*β*. In addition, the antioxidant function can be achieved by upregulating superoxide dismutase activity. The results suggest that chrysanthemum stems and leaves have a similar effect on intestinal inflammation as chrysanthemum. The anti-inflammatory effect of water extract of the chrysanthemum stem and leaf group increased with the decrease in concentration, while the anti-inflammatory effect of the alcohol extract group was the best at medium concentration, and the antioxidant effect of both extracts was good at high concentration.

As a non-medicinal part of chrysanthemum, the yield of chrysanthemum stem and leaf was significantly higher than that of chrysanthemum, but it could not be effectively used, resulting in resource waste. Zhou et al. used the HPLC-MS method to identify the chemical constituents of Chrysanthemum chrysanthemum, and identified 14 chemical components, including Chlorogenic acid, Apigenin-7-O-Glu, Diosmetin-7-glucoside, Luteolin-7-*O*-*β*-D-glucoside, Isochlorogenic acid A, Luteolin, Apigenin, Hesperidin, etc. [38]. Lin et al. identified phenolic compounds in methanol extract of Flos chrysanthemum by liquid chromatography with a diode array and electrospray ionization/mass spectrometry (LC–DAD–ESI/MS), fifteen caffeoylquinic acids and 15 flavonoids were positively identified, and the remaining compounds were provisionally identified, including Acacetin, Caffeic acid, and 1,3-Dicaffeoylquinic acid [39]. In this study, a UPLC–TQ/MS method for the determination of water and alcohol extracts from chrysanthemum stem and leaves was established and verified. The results show that the method is accurate and reliable. Suitable for the chrysanthemum leaf of Neochlorogenic acid, Chlorogenic acid, Caffeic acid, Cryptochlorogenic acid, 1,3-Dicaffeoylquinic acid, Isochlorogenic acid B, Isochlorogenic acid A, Isochlorogenic acid C, eight kinds of phenolic acids and Luteolin-7-*O*-*β*-D-glucoside, Apigenin-7-*O*-*β*-D-glucoside, Diosmetin-7-glucoside, Eriodictyol, Linarin, Tilianin, Apigenin, Diosmetin, Hesperetin, Acacetin, Luteolin and 11 kinds of flavonoids determination. In addition, neutral and acidic polysaccharides were determined by the phenol–sulfuric acid method and carbazol–sulfuric acid method, and the total polysaccharide content in chrysanthemum stems and leaves was obtained. The results showed that chrysanthemum stem and leaf contained similar components to chrysanthemum, among which the contents of total flavonoids and total phenolic acids in the alcohol extract of chrysanthemum stem and leaf were significantly higher than those in the water extract of chrysanthemum stem and leaf, but the water-soluble polysaccharides were significantly higher in the water extract of chrysanthemum stem and leaf.

In this present study, the experimental results showed that different extracts from chrysanthemum stem and leaf have certain anti-inflammatory and antioxidant effects, and the main bioactive components were analyzed, including flavonoids, phenolic acid, and polysaccharides. However, the anti-inflammatory and anti-oxidation mechanisms and targets of these bioactive components need to be further studied and explored.

## 4. Materials and Methods

### 4.1. Animal Care Ethics

All zebrafish experiments were conducted according to the guidelines of the Animal Ethics Committee of the Laboratory Animal Center, Nanjing University of Traditional Chinese Medicine.

### 4.2. Zebrafish Husbandry

The adult AB strain zebrafish (YSY Biotechnology Inc., Nanjing, China) were raised in a dark/light (10 h:14 h) [40], 26 °C environment, and live brine shrimp feed was cast twice a day. One day before the experiment, healthy zebrafish with reproductive ability were placed in spawning boxes with a 1:1 or 1:2 male to female ratio, separated by a partition. The partitions were pulled out at 9:00 the next morning, and the fertilized eggs were collected in the box at 11:30. The collected embryos were placed in a constant temperature incubator at 28 °C and kept under 14 h light. E3 culture water [41] (5 mM NaCl, 0.17 mm KCl, 0.33 mm CaCl_2_, 0.33 mm MgSO_4_, pH = 7.2) was prepared, and E3 was used as a specific medium for embryo culture, rather than zebrafish fish culture system water, to ensure stable buffer salt concentration and *p*H. Dead eggs were removed every 24 h, and fresh embryo culture medium was replaced.

### 4.3. Chemicals and Regents

The stems and leaves of chrysanthemum were collected from Yangma Town, Sheyang County, Yancheng City, Jiangsu Province in November 2019, and extracted and concentrated by Kanion Pharmaceutical using industrial percolation extraction, vacuum concentration, and other processes to obtain water extraction (ST) and 80% alcohol extraction (CT) extract respectively.

The control substance Isochlorogenic acid A, Isochlorogenic acid B, Isochlorogenic acid C, Cynarin (1,3-dicaffeoylquinic acid), Eriodictyol, Apigenin-7-*O*-*β*-D-glucoside, Cryptochlorogenic acid, Tilianin, Neochlorogenic acid, Diosmetin, and Diosmetin-7-glucoside were purchased from Shanghai Yuanye Bio-Technology Co., LTD (Shanghai, China). Caffeic acid, Chlorogenic acid, Luteolin, and Luteolin 7-*O*-*β*-D-glucoside, Apigenin, Acacetin, Linarin, and Hesperetin were purchased from the Nanjing chunqiu Biological Engineering Co., Ltd (Nanjing, China). The purity of the above 28 reference substances were more than 98% by HPLC. HPLC-grade acetonitrile and methanol were from Merk (Darmstadt, Germany). Other chemical reagents purchased from Shanghai Sinopharm Chemical Reagent Co., LTD (Shanghai, China), are analytical pure. Dextran sulfate sodium salt (DSS), molecular weight 36,000–50,000 (Lot Number: S3045, MP, Biomedicals, LLC). Glucuronic acid and glucose were purchased from Aladdin Biochemical Technology Co., LTD (Shanghai, China).

Deionized water was distilled and purified by an EPED super-purification system (Eped, Nanjing, China).

### 4.4. Apparatus

ACQUITYTM UPLC system, XevoTM TQ mass spectrometry system (Waters Corporation, Milford, MA, USA); Startorius BT1250 electronic balance (Sartorius Scientific Instruments Corporation, Beijing, China); TDL240B centrifuge (Anting Scientific Instrument Corporation, Shanghai, China), WH-1 micro-vortex mixing instrument (Shanghai, China); CENTRIVAP centrifuge enrichment apparatus (Labconco, Kansas City, MO, USA). EPED super-purification system (Eped, Nanjing, China).

### 4.5. Sample Preparation

A standard stock solution of 19 phenolic acids and flavonoids was prepared with methanol as solvent. They were 1.62 mg/mL for Neochlorogenic acid, 1.90 mg/mL for Chlorogenic acid, 1.86 mg/mL for Cryptochlorogenic acid, 1.46 mg/mL for 1,3-dicaffeoylquinic acid, 1.96 mg/mL for Isochlorogenic acid B, 1.72 mg/mL for Isochlorogenic acid A, 1.85 mg/mL for Isochlorogenic acid C, 1.74 mg/mL for Caffeic acid, 1.10 mg/mL for Luteolin-7-*O*-*β*-D-glucoside, 0.90 mg/mL for Apigenin-7-O- glucoside, 1.07 mg/mL for Diosmetin-7-glucoside, 1.10 mg/mL for Eriodictyol, 1.03 mg/mL for Linarin, 0.78 mg/mL for Tilianin, 1.29 mg/mL for Apigenin, 1.12 mg/mL for Diosmetin, 1.05 mg/mL for Hesperetin, 1.00 mg/mL for Acacetin, and 1.26 mg/mL for Luteolin.

Then, the appropriate amount of reserve solution of 8 phenolic acid and 11 flavonoid reference substances were taken separately, and methanol was added to make two mixed reference substance solutions containing each reference substance in the appropriate concentration range. The final concentration of phenolic acids were: 0.34–135.00 μg/mL for Neochlorogenic acid, 0.40–158.00 μg/mL for Chlorogenic acid, 0.39–157.00 μg/mL for Cryptochlorogenic acid, 0.31–122.00 μg/mL for 1,3-dicaffeoylquinic acid, 0.41–163.00 μg/mL for Isochlorogenic acid B, 0.36–143.00 μg/mL for Isochlorogenic acid A, 0.38–154.00 μg/mL for Isochlorogenic acid C, 0.36–145.00 μg/mL for Caffeic acid. The final concentration of flavonoids were: 0.17–69.00 μg/mL for Luteolin-7-*O*-*β*-D-glucoside, 0.20–81.00 μg/mL for Apigenin-7-*O*-glucoside, 0.22–89.00 μg/mL for Diosmetin-7-glucoside, 0.17–69.00 μg/mL for Eriodictyol, 0.16–64.00 μg/mL for Linarin, 0.12–49.00 μg/mL for Tilianin, 0.20–81.00 μg/mL for Apigenin, 0.18–70.00 μg/mL for Diosmetin, 0.16–66.00 μg/mL for Hesperetin, 0.16–62.50 μg/mL for Acacetin, and 0.20–79.00 μg/mL for Luteolin. A series of mass concentrations was prepared by successive dilution, which was used to examining the linear relationship. The mixed reference solution with different concentrations was filtered by 0.22 μm microporous membrane and stored at 4 °C for later use. An appropriate amount of glucose and glucuronic acid reference substance was taken, weighed accurately, and placed in a 10 mL volumetric flask. After dissolved in pure water, diluted to scale, and mixed, the glucose and glucuronic acid reference stock solutions with concentrations of 0.195 and 0.195 mg/mL were obtained and stored at 4 °C for later use.

Water extract and alcohol extract from industry were taken, and the yield was measured after freeze-drying. The freeze-dried yield of water extract and alcohol extract was 16.13% and 16.94% respectively. The water-extracted freeze-dried powder and alcohol-extracted freeze-dried powder were accurately weighed at about 1 mg each, and then placed in a 1 mL volumetric bottle, methanol was added to a constant volume to scale, and it was well mixed. It was then centrifuged at 13,000 r·min^−1^ for 15 min before injection, the supernatant was filtered through 0.22 μm microporous membrane, and the filtrate was used as the test solution and stored at 4 °C for later use.

### 4.6. Zebrafish IBD Modeling and Drug Treatment

Dextran Sulfate Sodium (DSS) is a sulfated polysaccharide that can cause chemical damage to the intestinal mucosa [35,36,37]. A zebrafish model with intestinal dysfunction was established using 0.5% DSS and the concentration was screened. The water extract and alcohol extract of chrysanthemum stem and leaf were prepared with different concentrations. According to the screening results, the concentration of alcohol extract from chrysanthemum stem and leaf was determined to be 0.02 mg/mL (CT0.02), 0.04 mg/mL (CT0.04), 0.06 mg/mL (CT0.06). The concentration of water was 0.2 mg/mL (ST0.2), 0.4 mg/mL (ST0.4) and 0.6 mg/mL (ST0.6). Then, 40–50 juvenile fishes on the third day post fertilization(dpf) were grouped into blank group, model group (immersed in 0.5% DSS), alcohol-extract group (immersion in different concentrations of alcohol extracts of chrysanthemum stems and leaves prepared with 0.5% DSS solution), and water-extract group (immersion in different concentrations of water extract solutions of chrysanthemum stems and leaves prepared with 0.5% DSS solution). Each group was repeated with 3 wells, the drug was administered for 72 h continuously, and the solution was changed every 24 h to observe the survival rate.

### 4.7. Histological Analysis

#### 4.7.1. Alcian Blue Staining

DSS-induced intestinal injury in zebrafish leads to accumulation of acidic mucin in the intestinal bulb. Moreover, a significant feature of DSS-modeled zebrafish is alcian blue staining in intestinal globules, and the accumulation of acidic mucin in intestinal globules is called “mucinous phenotype”. To observe the changes of intestinal goblet cells in the DSS model, alcian blue staining of acidic mucin was performed for comparison between the model group and the blank group [34]. Specific dyeing methods are as follows:(1)Fix the young fish in 4% paraformaldehyde at room temperature for 2 h.(2)Rinse the young fish with acidic ethanol (70% ethanol, 1% concentrated hydrochloric acid).(3)Soak young fish in Alcian blue staining solution (0.001% Alcian blue, 80% ethanol, 20% glacial acetic acid) for 15 min at room temperature.(4)Remove the staining solution and rinse the background repeatedly with acidic ethanol.(5)Place the young fish in 3% methylcellulose for imaging. Staining samples in acidic ethanol should be stored in sealed containers to avoid dehydration due to evaporation.

#### 4.7.2. Hematoxylin Eosin Staining

The 6 dpf zebrafish juvenile fish were immobilized overnight in 4% PFA at room temperature and then progressively dehydrated in ethanol. The juvenile fish were embedded in paraffin sections, cut into 3 μm sections, and stained with hematoxylin-eosin. Cross-sectional areas of the zebrafish intestine were scored using H&E staining and graded histologically. The analysis was modified as described in the literature and included grade of inflammation (0–3), degree of mucosal edema (0–3), degree of vacuolation (0–3), epithelial cell damage (0–3) and percentage of cupped cells (0–3) [42]. At least three juvenile fish were examined under each treatment condition.

### 4.8. Superoxide Dismutase (SOD)

SOD is a key enzyme in the antioxidant system, which can inhibit oxidative damage caused by external stimuli to the body, and the level of SOD can reflect the degree of the body’s anti-inflammatory response. Seventy-two hours later, 25 juvenile zebrafish in each group were collected and washed in a 1.5 mL centrifuge tube with normal saline 3 times and treated with 1:19 (zebrafish quality:normal saline volume). Then, 5% tissue homogenate was prepared by automatic sample rapid grinding machine, centrifuged at 4 °C for 5000 RPM × 15 min, and the supernatant was taken.

Before the test, 5% tissue homogenate was diluted into different concentrations for pre-experiment, and the concentration with an inhibition rate of 40–60% was selected. The pre-experiment results showed that the inhibition rate of 1% tissue homogenate met the requirements. The protein concentration of 1% tissue homogenate was determined according to the BCA method protein concentration determination kit. The activity of superoxide dismutase (SOD) in 1% tissue homogenate of each group was detected according to the instruction of the SOD assay kit.

Calculation formula:(1)SOD inhibition rate (%) = (A control − A control blank) − (A determination−A determination blank)/((A control − A control blank)) × 100%(2)SOD activity (U/mgprot) = SOD inhibition rate ÷50% × dilution ratio of reaction team system ÷ protein concentration of samples to be tested (mgprot/mL)

### 4.9. RNA Extraction and q-PCR

The 6 dpf zebrafish tissues were collected and placed into a homogenizer, and the RNA of each sample was extracted. The RNA of each group was reverse transcribed to obtain cDNA. The expression levels of genes related to inflammation were determined by the Bio-rad CFX96 real-time system. The results of β-actin were used as the internal reference. PCR primers of interleukin (IL) -1*β*, IL-8, Matrix metallopeptidase 9(MMP9), and reference gene *β*-actin were synthesized and purified by Shanghai Sangon Bio-Technology Co., Ltd (Shanghai, China), and their quality was detected. Primer information of RT-QPCR is shown in Table 2.

After the RT-QPCR reaction, the original data were derived. The original data were imported into Excel, and the 2^−(^^△△CT)^ calculation method was used for data processing. SPSS software was used to conduct one-way ANOVA, and *p* < 0.05 indicated a significant difference. GraphPad 8 software was used to draw experimental data pictures.

### 4.10. UPLC–TQ/MS Analysis

#### 4.10.1. UPLC-TQ/MS Conditions

A Waters Acquity UPLC BEH C_18_ column (2.1 × 100 mm^2^, 1.7 μm, Waters Corporation, Milford, Massachusetts state, USA) was used with a flow rate of 0.4 mL·min^−1^. The column temperature was 30 °C, the automatic sampler temperature was 10 °C, and the sample volume was 2 μL. The mobile phase gradient program was 0.1% formic acid water (A) and acetonitrile (B): 0–3 min, 4–7% B; 3–6 min, 7% B; 6–7 min, 7–15% B; 7–14 min, 15% B; 14–17 min, 15–35% B; 17–21 min, 35–100% B.

Electrospray ion source (ESI) was used. Simultaneous detection of positive and negative ions; IonSpray Voltage in positive and negative ion modes: 5500/−4500 V; ion source atomization temperature: 400 °C; curtain Gas: 30 psi; spray gas (Ion Source Gas1): 40 psi; auxiliary heating gas (Ion Source Gas2): 40 psi; scanning method: multiple reactive ion monitoring (MRM).

#### 4.10.2. Linearity

The preparation solution of mixed reference substance was diluted to prepare mixed reference substance solution with a series of mass concentrations. The retention time of each component was obtained from the mixed standards as a way to identify the components in the extracts. UPLC–TQ/MS analysis was performed according to the chromatographic conditions and the mass spectrometric conditions. With the peak area as Y and the concentration of reference solution as X, linear regression analysis was performed to calculate the correlation coefficient.

#### 4.10.3. Precision and Accuracy

Precision test: the mixed reference solution was injected 6 times in one day to determine the peak area of each component to be measured, and the intraday precision was evaluated by the relative standard deviation (RSD%) of each peak area.

Repeatability test: the last 6 samples of freeze-dried powder, each about 1.0 mg, were weighed accurately, prepared for the test solution, and the content and RSD value of each component to be measured in the test solution after UPLC–TQ/MS analysis were calculated.

Stability test: a sample solution from the repeatability test was injected into the liquid chromatography at 0, 2, 4, 8, 12, and 24 h, separately. After the UPLC–TQ/MS analysis, the peak area and RSD values of each component to be measured were calculated.

### 4.11. Determination of Total Polysaccharide Content

Using glucose and glucuronic acid as control, the neutral polysaccharide content in chrysanthemum stem and leaf was determined by the phenol–sulfuric acid method, and the acid polysaccharide content in chrysanthemum stem and leaf were determined by the carbazol–sulfuric acid method. Total polysaccharide content = neutral polysaccharide content + acid polysaccharide content.

### 4.12. Statistical Evaluation

MassLynx4.2 (Waters Corporation, Milford, Massachusetts state, USA) software was used to analyze the contents of total flavonoids and phenolic acids in water extract and alcohol extract of chrysanthemum stem and leaf, and SPSS22.0 was used for statistical analysis. Independent sample T test was used for comparison. A value of *p* < 0.05 was considered significant.

## 5. Conclusions

In summary, our study confirmed the anti-inflammatory and antioxidant effects of *Chrysanthemum morifolium* leaf extract in a zebrafish inflammatory bowel disease model and verified that its flavonoid, phenolic acid, and polysaccharide components were closely related to both anti-inflammatory and antioxidant activities. This study may also be helpful to develop the therapeutic agents for IBD chrysanthemum stem and leaf resources.

## Figures and Tables

**Figure 1 molecules-27-02114-f001:**
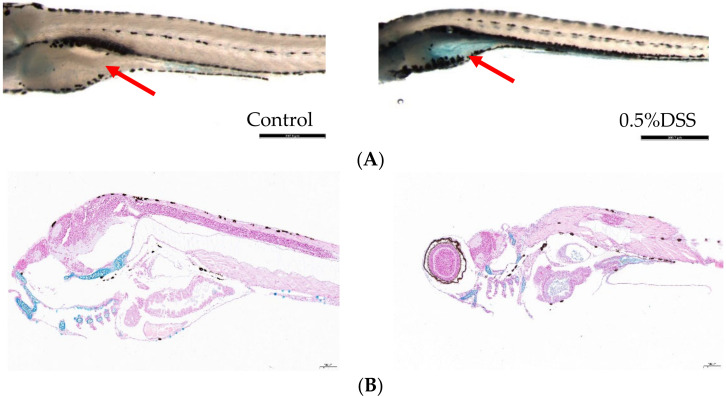
Alcian blue staining of zebrafish in blank group and 0.5% DSS model group. (**A**) Whole-mount control and DSS-exposed larvae stained with alcian blue. Red arrows indicate intestinal bulb staining, scale bar: 380 μm. (**B**) Alcian blue staining of longitudinal sections of juvenile fish from the blank and DSS-induced groups, scale bar 100 μm.

**Figure 2 molecules-27-02114-f002:**
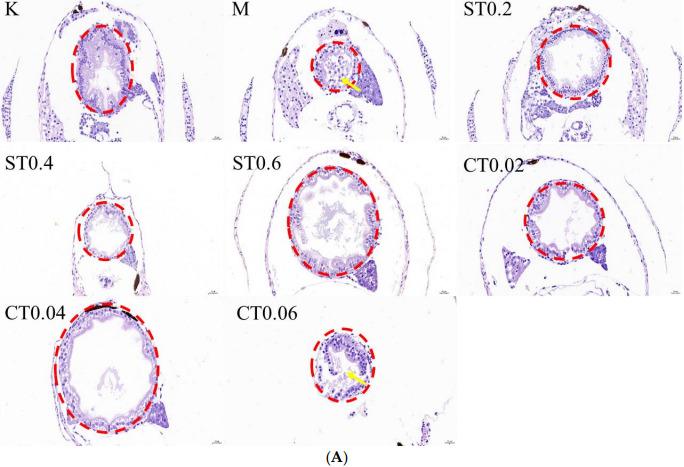
HE staining results and the proportion of intestinal lumen area in cross-sectional sections of zebrafish intestine in the blank, model and dosing groups. (**A**) HE staining map of zebrafish intestine. The red circle indicates the zebrafish gut section, and the yellow arrow indicates the broken intestinal epithelium. (**B**) Results of zebrafish intestinal HE staining score (*n* = 3). (**C**) The proportion of intestinal lumen area occupied by zebrafish. K: blank control group; M: model group; ST (0.2, 0.4, 0.6): water extract from chrysanthemum stem and leaf (0.2, 0.4, 0.6 mg/mL); CT (0.02, 0.04, 0.06): alcohol extract from chrysanthemum stem and leaf (0.02, 0.04, 0.06 mg/mL). (Model group vs. control group: ^###^ *p* < 0.001; drug administration group vs. model group: * *p* < 0.05, ** *p* < 0.01).

**Figure 3 molecules-27-02114-f003:**
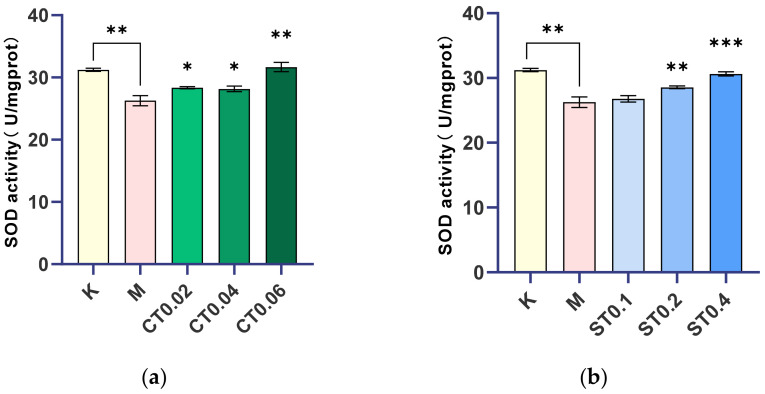
Comparison of SOD activity between blank group, model group, and chrysanthemum stem and leaf administration group. (**a**) Changes of SOD activity after modeling and alcohol extract of chrysanthemum stem and leaf were administered. (**b**) Changes of SOD activity after modeling and administration of water extract from chrysanthemum stem and leaf. K: blank control group; M: model group; ST (0.1, 0.2, 0.4): water extract from chrysanthemum stem and leaf (0.1, 0.2, 0.4 mg/mL); CT (0.02, 0.04, 0.06): alcohol extract from chrysanthemum stem and leaf (0.02, 0.04, 0.06 mg/mL). (* *p* < 0.05, ** *p* < 0.01, *** *p* < 0.001).

**Figure 4 molecules-27-02114-f004:**
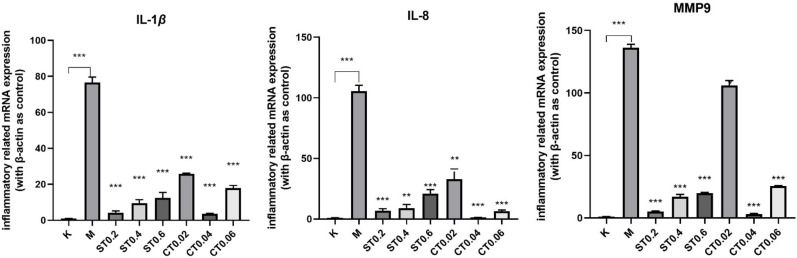
mRNA expression results of IL-1β, IL-8, MMP9 in the blank group, model group, and administration group. Compared with model group, ** *p* < 0.01, *** *p* < 0.001.

**Figure 5 molecules-27-02114-f005:**
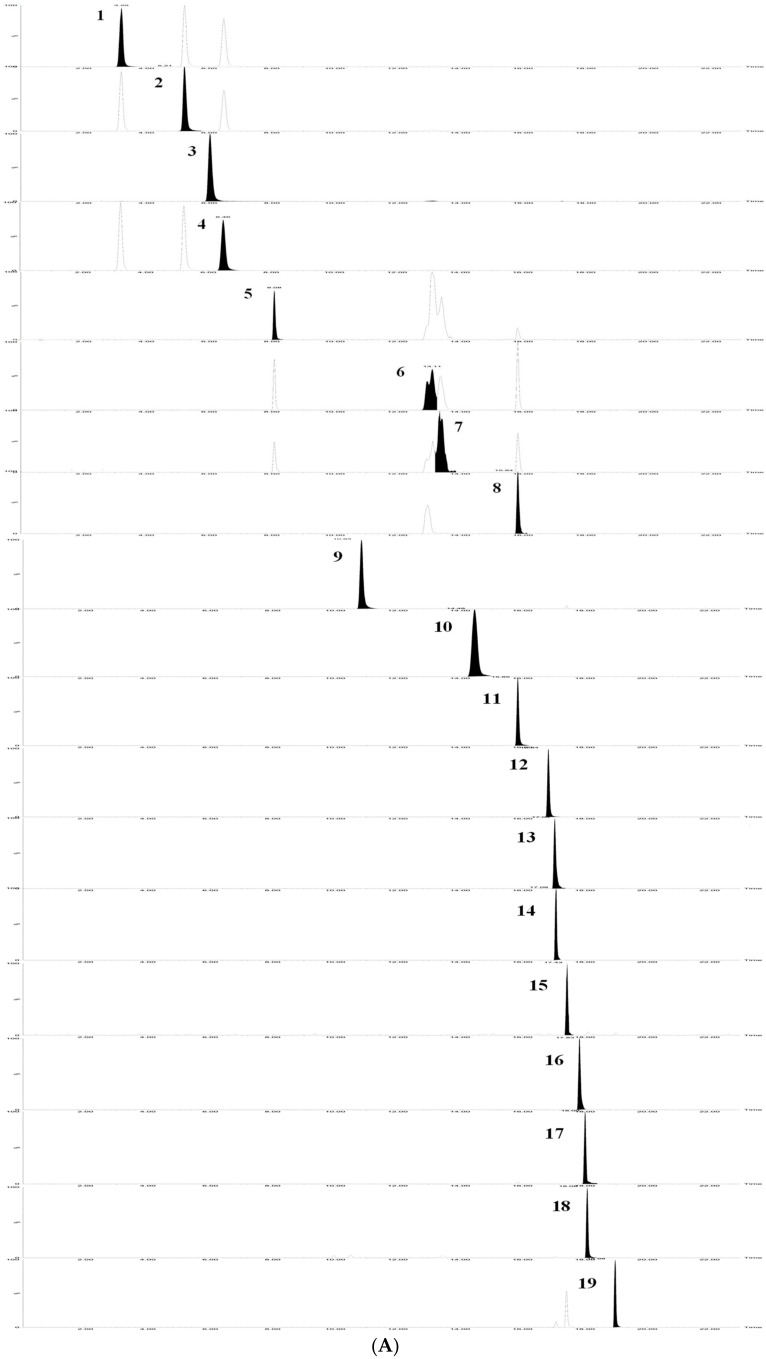
TQ chromatograms of the components to be measured in the mixed standards and the components to be measured in the samples. (**A**) TQ chromatogram of the components to be measured in the mixed standards. (**B**) TQ chromatogram of the components to be measured in the sample. According to the peak order, the components are respectively: (1) Neochlorogenic acid; (2) Chlorogenic acid; (3) Caffeic acid; (4) Cryptochlorogenic acid; (5) 1,3-Dicaffeoylquinic acid; (6) Isochlorogenic acid B; (7) Isochlorogenic acid A; (8) Isochlorogenic acid C; (9) Luteolin-7-*O*-*β*-D-glucoside; (10) Apigenin-7-O-β-D-glucoside; (11) Diosmetin-7-glucoside; (12) Eriodictyol; (13) Luteolin; (14) Linarin; (15) Tilianin; (16) Apigenin; (17) Diosmetin; (18) Hesperidin; (19) Actinin.

**Figure 6 molecules-27-02114-f006:**
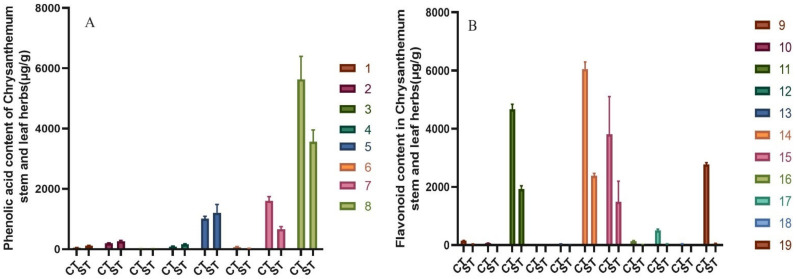
Contents of total phenolic acids and total flavonoids in alcohol extract from chrysanthemum stem and leaf (CT) and water extract from chrysanthemum stem and leaf (ST) of chrysanthemum stem and leaf. (*n* = 10). (**A**) Comparison graph of phenolic acid content in the extract; (**B**) Comparison graph of flavonoid content in the extract. According to the peak order, the components are respectively: (1) Neochlorogenic acid; (2) Chlorogenic acid; (3) Caffeic acid; (4) Cryptochlorogenic acid; (5) 1,3-Dicaffeoylquinic acid; (6) Isochlorogenic acid B; (7) Isochlorogenic acid A; (8) Isochlorogenic acid C; (9) Luteolin-7-*O*-*β*-D-glucoside; (10) Apigenin-7-O-β-D-glucoside; (11) Diosmetin-7-glucoside; (12) Eriodictyol; (13) Luteolin; (14) Linarin; (15) Tilianin; (16) Apigenin; (17) Diosmetin; (18) Hesperidin; (19) Actinin.

**Table 1 molecules-27-02114-t001:** A linear relationship between peak area and concentration of reference substance.

Number	Analytes	Calibration Curves ^a^	R^2^	Linear Range (μg/mL)
1	Neochlorogenic acid	y = 20367x + 14658	0.9987	0.34~135.00
2	Chlorogenic acid	y = 14377x + 12830	0.9975	0.40~158.00
3	Caffeic acid	y = 11510x − 359.18	0.9928	0.36~145.00
4	Cryptochlorogenic acid	y = 12772x + 15468	0.9993	0.39~157.00
5	1,3-dicaffeoylquinic acid	y = 44.539x + 37.236	0.9967	0.31~122.00
6	Isochlorogenic acid B	y = 1162.6x + 130.06	0.9999	0.41~163.00
7	Isochlorogenic acid A	y = 41.259x − 9.1657	0.9998	0.36~143.00
8	Isochlorogenic acid C	y = 23.482x + 28.95	0.9978	0.38~154.00
9	Luteolin-7-O-β-D-glucoside	y = 20728x + 7525.5	0.999	0.17~69.00
10	Apigenin-7-O-β-D-glucoside	y = 28985x + 2386.8	1	0.20~81.00
11	Diosmetin-7-glucoside	y = 10955x + 5588.4	0.9994	0.22~89.00
12	Eriodictyol	y = 35603x − 1415.7	0.9991	0.17~69.00
13	Luteolin	y = 290.93x − 4.1891	0.998	0.20~79.00
14	Linarin	y = 7679.9x + 2816	0.9986	0.16~64.00
15	Tilianin	y = 3.7865x + 3.4176	0.9979	0.12~49.00
16	Apigenin	y = 348.4x − 6.4809	0.9997	0.20~81.00
17	Diosmetin	y = 561.49x − 11.501	0.9961	0.18~70.00
18	Hesperetin	y = 90914x − 3254.3	0.9986	0.16~66.00
19	Acacetin	y = 28421x + 8227.3	0.9995	0.15~62.50

^a^ y is the value of peak area, and x is the value of the reference compound’s concentration (μg/mL).

**Table 2 molecules-27-02114-t002:** Primer information of RT-qPCR.

Gene	Forward Primer Sequence	Reverse Primer Sequence
IL-8	GTCGCTGCATTGAAACAGAA	CTTAACCCATGGAGCAGAGG
MMP9	CTTCAAGGACGGGCGCTACT	GAGGTGGTCCTCAAAGGCAG
IL-1*β*	TGGACTTCGCAGCACAAAATG	GTTCACTTCACGCTCTTGGATG
*β*-actin	CACACCGTGCCCATCTATGA	TTCTCTTTCGGCTGTGGTGG

## Data Availability

The data presented in this study are available on request from the corresponding author.

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
