# Peer review of "Evaluation of Anti-Inflammatory and Antioxidant Effectsof Chrysanthemum Stem and Leaf Extract on Zebrafish Inflammatory Bowel Disease Model"

_molecules, 2022, doi:10.3390/molecules27072114_

Round 1

Reviewer 1 Report

Dear authors,

The manuscript deals with an interesting topic and the objectives are stated clearly. However, this manuscript is not ready for publication yet. There are some concerns/issues that you need to address. I have uploaded the comments in the designated space. 

Thank you.

Reviewer 2 Report

The manuscript presents some non-compliance with the journal’s Instructions for authors. For example, the abstract has 303 words instead of 200 words maximum. In addition, abbreviations used in the text are not defined at the right site, for example, DSS. The title of your manuscript should be more concise.
Some Figures and Tables do not have a short explanatory title and caption; they must be expanded. For example, Figures 3, 4, 5, and 6 are confusing, and the Y-axis title cannot be read. In Table 1, please describe the reference substance.
In general, the manuscript must be considerably improved. The authors must pay attention to the Tables and Figures edition and the journal’s Instructions. They must also expand the description and interpretation of the experimental results and the conclusions drawn from the results presented.

Author Response

Thank you for your correction. We have modified the summary section and marked it in red. See lines 15-31 on page 1. We added the definition of DSS. See line 81 on page 2. We revised the title of the paper. We have modified Figures 3, 4, 5, and 6.

Reviewer 3 Report

The document evaluation of anti-inflammatory and antioxidant of chrysanthemum stem and leaf extract on zebrafish inflammatory bowel disease model and quantitation of the bioactive compositions by UPLC-TQ/MS is an interesting document of biological activity from phytochemical compounds evaluated in “in vivo” systems, nevertheless the information presented is not clearly described. Some comments about methodologies and results showed in the document need to be addressed to be considered for publication.

The objective needs to be clearly specified to follow a sequential order of activities, as well as the experimental analysis.

The quality of figure 3 must be improved, it is not possible to observe the title on the y axis. Similar for figure 4.

Information of linear relationship between peak and concentration or reference substances presented in table 1 is not clear, are they commercial standard compounds?  If those compounds are evaluated to indicate phenolic compounds concentrations, it is not necessary to present the in the body of the document and this information can be move to supplementary data, similar with table 2, as the principal objective of this document is not the development of the analysis technique of phenolic compound quantification.

Figure 5 is not clear, the information presented is the mass spectra of the standard mixture or the compounds presented in the extracts? How the identification of compounds presented in the extract was performed? Please explain in materials and section method and present the mass spectra of the compounds presented in the extracts.

The presentation of the data of acid phenolic and flavonoids must be improved, the figure 6 could indicate the concentration of the compounds with the error bars instead of the proportions, and the concentration of each tested system and the statistical analysis must be added.

How is possible to differentiate the activity of the phenolic compounds from the polysaccharide activity?

Statistical analysis is not indicated in materials and method section

Round 2

Reviewer 2 Report

The manuscript has been improved. However, in Table 1, the reference substance has not been described as solicited. Some abbreviations used in the text are not defined at the right site, for example, SOD, TCM, ST, CT, and TQ. The conclusions section has not been expanded as required. In general, the authors must pay attention to the Tables and Figures edition. 

Reviewer 3 Report

Table 1 could be moved to supplementary data, also part A of figure 5. Part B of figure 5 could be presented in summary form